# Populating Memory in Continual Learning with Consistency Aware Sampling

## Abstract

Continual Learning (CL) methods aim to mitigate Catastrophic Forgetting (CF), where knowledge from previously learned tasks is often lost in favor of the new one. Among those algorithms, some have shown the relevance of keeping a rehearsal buffer with previously seen examples, referred to as *memory*. Yet, despite their popularity, limited research has been done to understand which elements are more beneficial to store in memory. It is common for this memory to be populated through random sampling, with little guiding principles that may aid in retaining prior knowledge. In this paper, and consistent with previous work, we found that some storage policies behave similarly given a certain memory size or compute budget, but when these constraints are relevant, results differ considerably. Based on these insights, we propose CAWS (Consistency AWare Sampling), an original storage policy that leverages a learning consistency score (C-Score) to populate the memory with elements that are *easy to learn* and *representative* of previous tasks. Because of the impracticality of directly using the C-Score in CL, we propose more feasible and efficient proxies to calculate the score that yield state-of-the-art results on CIFAR-100 and Tiny Imagenet.

## 1 Introduction

Deep Learning models have repeatedly shown state of the art performance in numerous tasks, including image recognitionHe et al. (2016); Dosovitskiy et al. (2020), Natural Language Processing (NLP) Devlin et al. (2018); Brown et al. (2020) or games previously thought to be intractable to solve, such as Go Silver et al. (2016) and Starcraft II Vinyals et al. (2019). However, as a common limitation, all these models lack versatility: when trained to perform novel tasks, they rapidly forget how to solve previous ones. This condition is known as *catastrophic forgetting* and is the main problem tackled by Continual Learning methods Parisi et al. (2019); Delange et al. (2021).

A variety of methods have been proposed to approach this problem. Some have focused on allocating parameters sub-spaces for each new task Rusu et al. (2016); Mallya et al. (2018), others define restrictions on gradients learned Kirkpatrick et al. (2017); Lopez-Paz & Ranzato (2017), while others use meta-learning to learn reusable weights for all tasks Rajasegaran et al. (2020); Hurtado et al. (2021). Among these, memory-based methods like Experience Replay Chaudhry et al. (2019); Kim et al. (2020) have consistently exhibited greater performance while being easy to understand. In these methods, a memory of samples from previous tasks is kept during training of the current task to avoid forgetting how to solve previous tasks. Notwithstanding the popularity and effectiveness of memory-based methods, few studies have been conducted on *how populating the memory* affects the performance of CL methods. In particular, Chaudhry et al. (2018a); Wu et al. (2019); Hayes et al. (2020); Araujo et al. (2022) show that when populating the memory by focusing solely on sample diversity or class balance, random selection of elements ends up performing nearly or just as well without adding extra computation.

It is clear that having a representative set of examples of the underlying distribution is critical for preserving previous knowledge. Ideally, one would like to save a large number of samples. Unfortunately, since saving large amounts of data results in computational overhead, we have to limit the memory size and choose which elements to keep. In this paper, we argue that this memory must satisfy two fundamental requirements in order to perform reliably. The first is to have elements that are easy to remember, or that the model can learn quickly. The second is to have elements that are a

suitable representation of the distributions of past experiences, having diversity but being careful to avoid outliers. We refer to these two ideas as *Fast Learning* and *Diversity*.

For the first requirement, we leverage a concept called *learning consistency* (Jiang et al., 2021) to measure how consistently a sample is learned by a given class of a models. Specifically, we populate the memory with elements with higher consistency values, which have also been shown to be the fastest to learn. Thus, sample efficiency is improved for memory samples. However, by selecting only those samples with the highest consistency values, the model learns only a limited set of patterns, reducing the diversity of samples stored in the memory and limiting how much of the decision boundary the model is capable of representing. To overcome this, we propose *Consistency AWare Sampling* (CAWS) as a new populating strategy that incorporates sampling from a broader group of high consistency elements. This new proposal adds diversity to the memory while allowing the model to learn a far more detailed decision boundary.

One of the limitations of the consistency score (C-Score) is that it requires training multiple models with the entire training distribution in order to find out how easy or difficult it is to train an example, which, in addition to being expensive, is impractical for CL. To mitigate this problem, we propose proxies to calculate the consistency of an example, achieving similar performance without the need of training multiple models on the entire training set beforehand. These proxies are not limited to CL scenarios, they can also be used in environments where these types of scores are commonly used, such as in Curriculum Learning (Bengio et al., 2009).

Thus, our contributions can be summarized as follows:

- Taking a step towards understanding how the memory should be populated based on the effectiveness-efficiency trade-off of the scenario.
- In Section 3, we propose a novel method - Consistency AWare Sampling (CAWS)- for populating the memory for Continual Learning based on the idea of learning consistency. This method equals or outperforms state-of-the-art memory selection methods.
- Since learning consistency requires trained models on the same training data to be estimated, in Section 4 we propose practical proxies that require no extra training and achieve similar results. Moreover, these proxies could be used for other scenarios where C-Scores are required, such as Curriculum Learning.

## 2 FAST LEARNING

Multiple ways to populate memory in CL have been proposed. However, few studies have explored when different approaches work better than others. Some studies have shown that, under certain conditions, there is no significant difference between the proposed methods, showing how limited our understanding of how replay strategies is. In this work, we will consider the following methods as baselines:

**(a) Reservoir.** A reservoir (Vitter, 1985) strategy allows sampling elements from a stream without knowing how many instances to expect. The method selects each sample with a probability $\frac{M}{N}$ where $N$ is the number of elements observed so far, and $M$ is the memory size. This way, it acts randomly to maintain a uniform sample from the already seen stream.

**(b) Class Balance.** As the name states, each class has an equal proportion of the buffer size (Chrysakis & Moens, 2020). We use a dynamic assignment, meaning that the memory is always complete. Samples of new classes replace instances of old classes to maintain equal distribution in the memory.

**(c) Task Balance.** Similar to Class Balance, but instead of an equal proportion of classes, the memory is divided by the number of tasks the sequence has (Lopez-Paz & Ranzato, 2017).

**(d) Mean of Features (MF).** Proposed by Rebuffi et al. (2017), it calculates an average class feature vector, based on the representations of the elements in memory for a given class. If the distance of the new vector to the corresponding class vector is smaller than the farthest in the memory, we replace the new example with the farthest one.

These methods perform rather similar to one another. Table 1 shows the mean accuracy over the sequence of tasks obtained by these baseline methods. MF behaves similarly to Reservoir, Class Balance, and Task Balance when we train each task for a limited amount of epochs (1). However, when increasing the number of epochs, we see that MF outperforms the other methods considerably, showing the importance of correctly populating the memory in some settings. The behavior is not limited to a simple environment, as it can be replicated when changing the memory size and the model used.

The similarity between methods when we have limited computation suggests that elements selected are unsuitable when training for a small number of epochs. Given the selected memory, the model cannot correctly identify relevant patterns from previous tasks. We hypothesize that: *when we face restrictions on the number of epochs that a model can train or in the memory size, we need to focus only on those elements that are faster to learn.*

Given this implication, and taking a page from the Curriculum Learning literature, we propose prioritizing samples that are faster to learn based on learning consistency. In particular, we propose to use a metric developed for Curriculum Learning called Consistency Score (Jiang et al., 2021). This metric has shown benefits for faster learning under compute restrictions while also providing more robust learning to noise (Wu et al., 2020) in Curriculum Learning.

Learning Consistency or C-Score measures how learnable a specific sample can be concerning a set of models. It has been proposed as a task agnostic measure of the difficulty used in Curriculum Learning. We can define the C-Score as follows: let $D$ be a dataset of size $n$ sampled from an underlying distribution $\mathcal{P}$. Let $f(\cdot, D)$ be a model trained on $D$. Then, for an instance $(x, y)$ of $D$, Learning Consistency is defined as:

$$\mathcal{C}_{\mathcal{P},n}(x, y) = \mathop{\mathbb{E}}_{D \overset{n}{\sim} \mathcal{P}} \left[ \mathbb{P}(f(x; D \setminus \{(x, y)\}) = y] \right] \tag{1}$$

In practice, estimating this score for each sample is computationally intractable, thus, approximations need to be made. Current approximations, however, still require training thousands of models over a training set to acquire the score of each sample. Thankfully, the authors of the original paper provide precomputed C-Scores for MNIST, CIFAR-10, and CIFAR-100. For other datasets, in contrast, we need more efficient proxies. In particular, in the same work, the authors propose to approximate the C-Score based on the learning speed of samples. However, this still requires registering the percentage of iterations where a sample has been correctly classified.

To test the importance of our hypothesis, we devise two different methods where we populate memory based on learning consistency. The first one is High C-Score (*High-C*), which samples only the top $N$ most consistent elements of each class. The second one is Low C-Score (*Low-C*), which, contrary to *High-C*, selects only elements with the lowest C-Score. This division helps us verify that selecting easier (*High-C*) or difficult-to-learn (*Low-C*) elements can strongly impact the sequence's final accuracy.

## 2.1 EXPERIMENTAL SETUP

To provide an empirical validation to our hypothesis, we focus on a Class-Incremental setting, as has been the main focus of recent Continual Learning scientific endeavors. Such scenario is much more challenging and more realistic than the traditional Task Incremental setting (Van de Ven & Tolias, 2019). In Class Incremental scenarios, each task $t$ consists of a new data distribution $D^t = (X^t, Y^t)$, where $X^t$ denotes the input instances and $Y^t$ denotes the instance labels. The goal is to train a classification model $f : X \longrightarrow Y$ using data from a sequence of $T$ tasks: $D = \{D^1, ..., D^T\}$. Each task is presented sequentially to the model and trained for $E$ epochs. Crucially, unlike the Task Incremental setting, *a task descriptor is only available during training*.

### 2.1.1 DATASETS

We train our models on different sequence of CIFAR-10 and CIFAR-100 (Krizhevsky & Hinton, 2009) splits in 5 tasks, and TinyImageNet (Le & Yang, 2015) split in 10 and 20 tasks. CIFAR10 and CIFAR-100 data-sets are traditionally used in Continual Learning and have the advantage of

Table 1: Accuracy for models trained with different memory population methods for 1, 5 and 10 training epochs. Selecting only from the most consistent samples improves accuracy when we have a limited budget (1 epoch). However, as we increase the number of epochs, we can see that Mean Features achieves better results. This behavior holds for different memory sizes and architectures.

| # E | CIFAR100-500 | | | CIFAR100-1000 | | | CIFAR100-500-RS | | |
|---|---|---|---|---|---|---|---|---|---|
| | 1 | 5 | 10 | 1 | 5 | 10 | 1 | 5 | 10 |
| Reservoir | 11.0% | 14.9% | 15.6% | 11.1% | 18.3% | 19.3% | 15.4% | 22.8% | 25.5% |
| Class Bal | 11.5% | 14.9% | 15.9% | 11.5% | 18.2% | 18.7% | 15.7% | 23.1% | 26.0% |
| Task Bal | 11.0% | 14.9% | 15.6% | 11.5% | 18.4% | 19.3% | 15.0% | 22.9% | 25.5% |
| MF | 12.2% | 16.4% | 17.7% | 11.7% | **20.9%** | 20.6% | 16.28% | **25.5%** | **28.0%** |
| Low-C | 6.9% | 9.9% | 11.3% | 7.0% | 9.7% | 11.4% | 9.0% | 14.0% | 16.2% |
| High-C | **14.5%** | **17.7%** | **18.6%** | **15.2%** | 19.9% | **20.8%** | **18.2%** | 25.3% | 27.3% |

having available precomputed C-Scores. On the other hand, for TinyImageNet, we compute the C-Score using the approximation proposed in (Jiang et al., 2021). These datasets also provide different distributions of C-Scores, with CIFAR-10 and TinyImageNet having highly skewed distributions, while CIFAR-100 shows a much more uniform distribution of C-Scores, as shown in Figure 8 in the Appendix.

### 2.1.2 Implementation Details

All experiments are run with 3 different seeds, each inducing a different ordering of sequences. In the case of CIFAR-10 and CIFAR-100, we use a simple convolutional architecture proposed in Mirzadeh et al. (2022). To better understand the behavior of our proposal, we also use a reduced Resnet-18 (RS) (Rebuffi et al., 2017) for CIFAR-100. This last model is also used for Tiny-Imagenet experiments. The optimizer is SGD with a learning rate of $0.001$, momentum $0.9$, and batch size $32$, unless otherwise mentioned. All methods are trained using Avalanche (Lomonaco et al., 2021), and the proposed methods' plugin will be released and integrated in the library upon acceptance. As proposed by Lopez-Paz & Ranzato (2017), we use the average performance over the T tasks after the sequential learning (Acc), and the forgetting (For) measured by how much performance is lost on previous tasks after sequential learning. Equation 2 shows the formulas for the metrics, where $A_{i,j}$ is the accuracy of task $i$ after training task $j$.

$$Acc = \frac{1}{T} \sum_{i=1}^{T} Acc_{T,i} \qquad For = \frac{1}{T-1} \sum_{i=1}^{T-1} Acc_{T,i} - Acc_{i,i} \qquad (2)$$

### 2.2 Results

Since we are populating the memory with only the most consistent elements, we expect accuracy to increase when the number of epochs is limited. Looking at the results in Table 1, we can indeed confirm this overall trend. These results occur not only with different memory sizes and models in CIFAR-100, as indicated in the columns of Table 1, but also in CIFAR-10, as shown in Figure 1. Despite highly increasing the memory size in CIFAR-10, the effect on accuracy is still considerable when we train for a single epoch. Focusing on Figure 1b, the primary reason for the increase in accuracy is the mitigation of forgetting when populating with the High-C strategy. On the other hand, we see that the opposite effect happens when we populating the memory with only the least consistent samples. These results are expected, as these are the most difficult elements to learn.

We hypothesize that highly consistent samples are easily classified because they represent common patterns in the dataset. Thus, training with those samples rapidly reinforces learning patterns that can be applied to a broader set of samples from the same dataset, increasing the learning speed. On the other hand, highly inconsistent samples would depend on patterns that are specific to fewer samples or outliers. While High-C samples common patterns, it still represents a limited spectrum of total dataset patterns, which limits its ability to remember past experiences. We find supporting evidence for this hypothesis in Figure 1c. In it, we see that increasing the number of epochs used

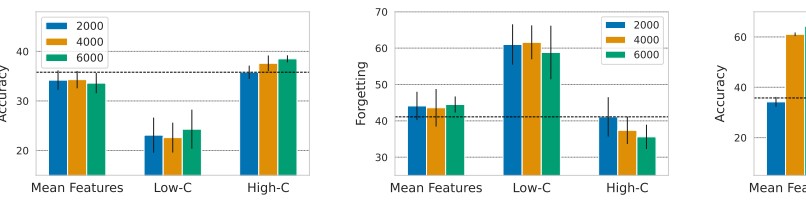

(a) Accuracy vs Memory Size  (b) Forgetting vs Memory Size  (c) Accuracy vs Training Epochs

Figure 1: Different metrics for Mean of Features, Low-C, and High-C methods when trained on CIFAR-10. Figure (a) shows that when increasing sample diversity by increasing the memory size, one can observe that populating with High-C values can improve performance when training for 1 epoch. These results are explained because High-C can better mitigate forgetting. On the other hand, by keeping the memory size at 2000 but changing the number of epochs, the performance of High-C almost does not change, showing the limitation of the method.

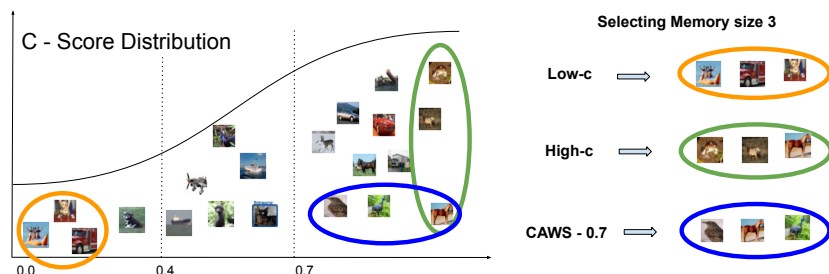

Figure 2: Difference in how the evaluated methods select samples from different sections of the C-Score distribution for a memory of size = 3. *Low-c* only selects elements with the lowest C-Score. *High-c* selects only those with the highest C-Score. On the other hand, *CAWS* samples from a sub-group of elements with C-Scores higher than a given threshold, selecting randomly from this group.

in each task does not result in better performance when training on highly consistent samples as in other methods. The low diversity from selecting only the highest consistency elements negatively affects the performance since the memory cannot fully represent previous distributions.

## 3 DIVERSITY

As noted in the previous section, by only selecting the most consistent elements, the model can learn effectively when given a few epochs per task. However, it can suffer from a lack of diversity when training for more epochs. To avoid this problem, and as part of our current hypothesis, we want to increase the diversity of the samples in memory.

With previous results, we had shown that selecting samples that are easy to learn can help during training. At the same time, we learn to avoid less consistent elements since these tend to represent outliers of each task. Given these insights, we propose Consistency Aware Sampling (CAWS), which populates the memory by randomly selecting from the top $X\%$ of C-Score samples. We call this percentage the **Sampling Ratio** of CAWS.

Both the *High-C* and *Low-C* scenarios emphasize learning consistency without diversity as they only have access to a smaller range of input values. On the other hand, CAWS gets to choose from a pool of high learning consistency samples over a wider spread of the training distribution. In Figure 2, there is a visual explanation of how CAWS populate the memory. In the Appendix (Algorithm 1), an algorithm explaining the procedure.

### 3.1 RESULTS

One of the problems we had with selecting only the most consistent elements was that the model quickly forgot when we increased the number of epochs. Unlike High-C, CAWS can better repre-

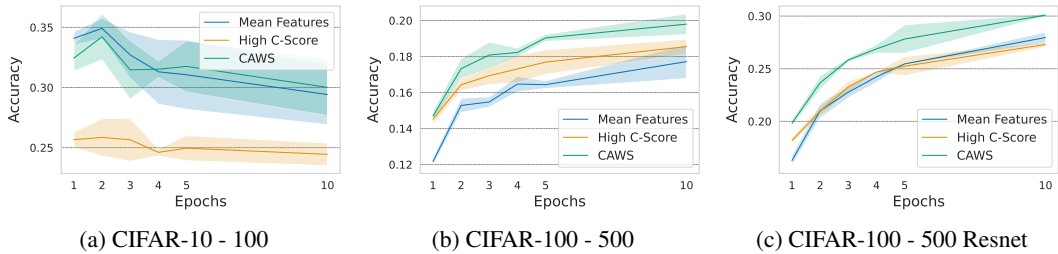

(a) CIFAR-10 - 100      (b) CIFAR-100 - 500      (c) CIFAR-100 - 500 Resnet

Figure 3: Model accuracy for different values of training epochs per task. Models trained on CIFAR-10 and CIFAR-100 with a memory size of 100 and 500, respectively. Only selecting samples with high C-Score does not work as well as other methods. However, when mixed with random selection, as with CAWS, it equals or outperforms the baselines.

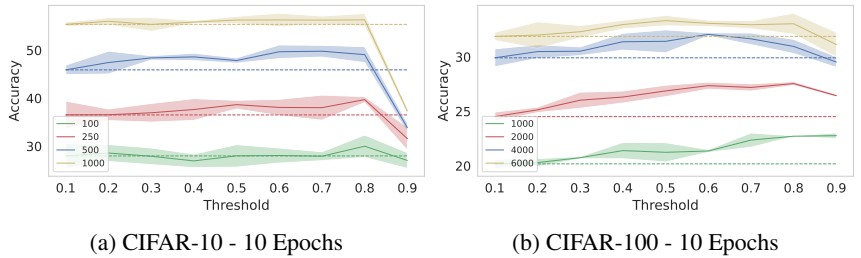

(a) CIFAR-10 - 10 Epochs      (b) CIFAR-100 - 10 Epochs

Figure 4: Accuracy for models trained using CAWS for different sampling ratios. Accuracy behaves as an inverted U-shape between two extremes. When the sampling ratio is 0.0, behavior is the same as randomly selecting elements with maximal diversity. When close to 1.0 we are choosing a small selection of the training data that is easy to learn but with little diversity. In between, there is an optimal point where diversity and rapid learning balance each other.

sent previous tasks in memory, achieving better accuracy and less forgetting in different scenarios. Figures 3b and 3c indicate that CAWS can add a better representation of previous experiences to memory, achieve even better accuracy than MF when training for multiple epochs.

The compactness of some datasets' C-Score distribution makes it difficult for High-C to represent them with only the most consistent elements. This effect does not occur as drastically in datasets where the distribution of the C-score is more uniform. For scenarios where High-C is inefficient, CAWS performs similarly to the previous methods when adding diversity, as shown in Figure 3a.

## 3.2 UNDERSTANDING THE IMPORTANCE OF SAMPLING RATIO

Figure 4 shows how altering the sampling ratio affects the performance of CAWS. The performance moves in an inverted U-Shape. When close to 0.0 it is the same as randomly selecting memory elements. Ideally, this is where our memory may have access to samples with the greatest diversity. However, this diversity comes with the price of being costly to learn in both compute and sample efficiency. When we move closer to 1.0, we get efficient samples but lack enough diversity to model the decision boundaries of the classifier accurately. We observe that a sampling ratio between 0.6 and 0.8 produces the best results in different datasets, where a balance of sample diversity and learning speed is attained.

One can observe the relationship between memory size and the behavior of *Sampling Ratio* to study further the diversity obtained using CAWS. Figures 4 shows that when the memory capacity increases, the optimal value tends to decrease, which agrees with our CAWS diversity hypothesis. When the memory capacity is low, it is preferable to have easy-to-learn examples that can correctly represent a small group of data from the previous tasks. On the other hand, when memory grows, a lower *Sampling Ratio* helps increase diversity, improving the representativeness of multiple patterns. These results imply that if memory is limited, it is better to have less variety but well-represented patterns than to have poorly represented patterns with high diversity.

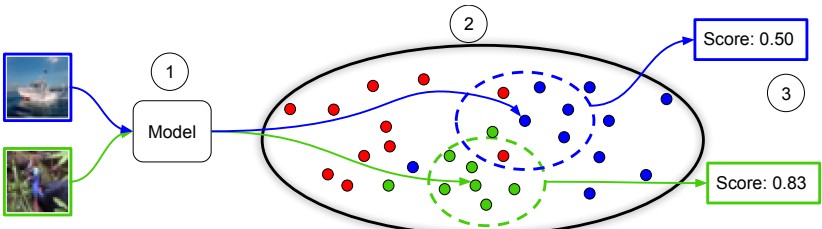

Figure 5: To get the proposed proxies for the consistency score, first (1) we must obtain a feature vector for each sample of the dataset. These vectors come from a pre-trained model or the current model depending on the proxy used. Then, for each sample, the $N$ nearest neighbors (2) are retrieved to count how many of those belong to the same sample class (3).

## 3.3 LIMITATIONS OF USING C-SCORES

Though CAWS outperforms other memory population methods, we understand that it requires having access to C-Scores, which are highly costly to compute. Moreover, they require training models on the same training distribution one hopes to learn. This assumption does not hold in the Continual Learning scenario. Thus, we propose a few proxies for the C-Score, which perform similarly without any of its current limitations. The following section describes how they work.

## 4 C-SCORE PROXIES

To develop proxies for the C-Score, we base ourselves on the Critical Sample Ratio (CSR) from Arplt et al. (2017) and the relative local-density scores proposed in Jiang et al. (2021). These metrics relate fast learning with the relation of a given sample with neighboring samples in feature space. The idea is that harder-to-learn decision surfaces have a greater mixture of samples with different labels than easier ones. They find that pairwise distance proxies work in feature space after having a model trained on the training distribution. These findings, of course, are impractical. However, when using a pre-trained model, one might get a reasonable estimate of the C-Score without training on the training data. Thus, our proxy consists of counting the ratio of neighbors of a given sample that are from the same class:

$$\hat{C}^L(x, y) = \frac{1}{N} \sum_{i=1}^{N} \mathbf{1}[y = y_i]$$

Where $\{(x_1, y_1), (x_2, y_2), ..., (x_n, y_N)\}$ are the closest $N$ neighbours of sample $(x, y)$ in the pre-trained model's feature space. We use cosine distance as our distance metric and test using different values of $N$. Empirically we found that between 5 and 200 it shows similar results, so we decided to use a value of N equal to 100.

We test 3 progressively easier to calculate versions of the proxy in our experiments:

- **Proxy 1:** using data of the complete sequence, we use a pre-trained model to calculate the embedding of each sample. Then we calculate neighbors. This proxy is still impractical for the Continual Learning Scenario but can be used as a strong baseline.

- **Proxy 2:** using data only from the current task and using a pre-trained model, we calculate the neighbors of the samples of the current task.

- **Proxy 3:** we use data only from the current task to calculate neighbours but use the *current model's* embeddings.

For simplicity, the pre-trained model used is a ResNet-18 with the weights obtained from PyTorch, but even models pre-trained in a self-supervised way can be used.

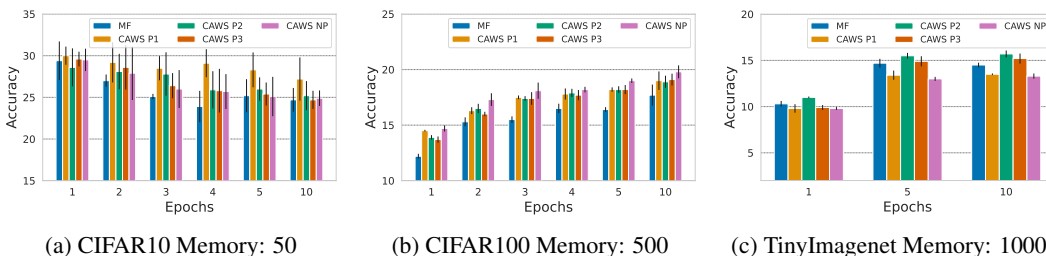

(a) CIFAR10 Memory: 50     (b) CIFAR100 Memory: 500     (c) TinyImagenet Memory: 1000

Figure 6: Accuracy for models trained with CAWS using increasingly more relaxed versions of proxy C-Scores compared against the baseline Mean Features method for different number of training epochs. As can be seen, CAWS using any proxy outperforms the baseline significantly. Performance differences between proxies are far less significant suggesting they can indeed be practically used for Continual Learning Scenarios. NP: No Proxy.

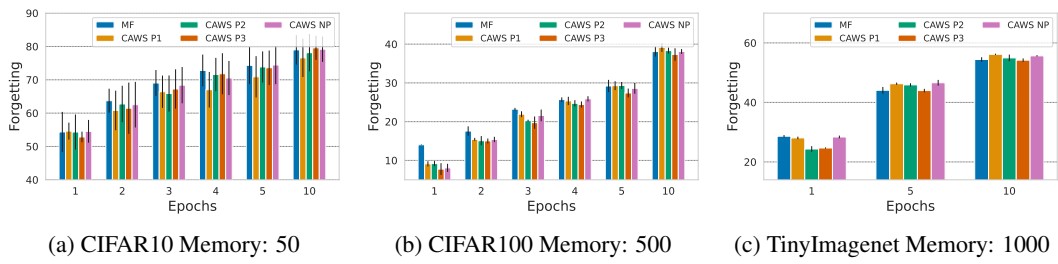

(a) CIFAR10 Memory: 50     (b) CIFAR100 Memory: 500     (c) TinyImagenet Memory: 1000

Figure 7: Similar to Figure 6 but with Forgetting. In most cases, forgetting decreases when we approximate the C-Score. The results agree with section 1, where forgetting is much lower in our proposal than MF, when training for fewer epochs. Although the gap narrowed as we increased the epochs, Forgetting is still lower in CAWS than in MF.

## 4.1 RESULTS

We compare the Mean Features baseline with the different proxies of CAWS in Figure 6. The first thing that stands out is that proxies continue outperforming MF. By utilizing our approximations, we achieved a value that can be compared to the consistency of each task's samples. However, a direct comparison between the consistency values is not possible since they represent different ways of calculating consistency. These results also suggest that CAWS can be used in practical scenarios for Continual Learning, as both Proxies 2 and 3's requirements are easy to meet in most applications.

Sometimes, proxy-based CAWS achieves better results than using the ground truth C-Scores. Two complementary reasons can explain this. The first is that these approaches can help improve memory diversity without weakening the previous tasks representativeness. Second, these approaches achieve a better consistency score than the C-Score. It is beyond the scope of this paper to verify this hypothesis, as more studies are needed, focusing on different scenarios, not only CL. Nonetheless, we consider it an interesting topic for future research.

Based on Figure 6c, we observe that CAWS with the ground truth C-Score is less effective than other simpler proxy methods in the case of Tiny ImageNet. In fact, its performance is even lower than MF. As previously noted, Tiny Imagenet's C-Score is approximated using an approximation proposed in the original paper (Jiang et al., 2021), which we suspect may not be as strong as its ground truth C-Score.

## 5 RELATED WORK

Memory-based methods mitigate CF by inserting data from previous tasks into the training process of the current one (Ebrahimi et al., 2021; Buzzega et al., 2021). These approaches can either use raw samples(Rebuffi et al., 2017; Chaudhry et al., 2019), minimize gradient interference (Lopez-Paz & Ranzato, 2017; Chaudhry et al., 2018b) or train generative models such as GANs or autoencoders

(Lesort et al., 2019; Shin et al., 2017; Kemker & Kanan, 2018) to generate elements from previously-seen distributions.

Multiples approaches has been propose to populate the memory. One simple but efficient method is the Reservoir strategy (Vitter, 1985), which randomly select the elements that go into the memory buffer. Other strategies have been proposed by adding different metrics to populate the memory with more representative elements (Chaudhry et al., 2019; Hayes et al., 2020; Hayes & Kanan, 2021; Aljundi et al., 2019b). Other works have focused in measuring the impact of hyperparameters on certain methods (Merlin et al., 2022), or studied the effect that rehearsal methods have on the loss functions (Verwimp et al., 2021). A different line of work has focused on how to select elements from the memory, either by how much the loss of an element is affected (Aljundi et al., 2019a) or by a ranking based on the importance of preserving prior knowledge (Isele & Cosgun, 2018).

Yet, in spite of the popularity of memory-based methods, little has been studied about the impact of memory composition on Continual Learning (Tiwari et al., 2022). Some proposals along this line are based on applying reservoir strategies (Chrysakis & Moens, 2020), while others have proposed to use entropy-based functions to increase memory diversity(Wiewel & Yang, 2021; Sun et al., 2021). Others have increased diversity by minimizing the angles of the gradients between different elements (Aljundi et al., 2019b). Despite improving performance in certain scenarios, few studies have been done targeting how to improve memory representativeness.

Other definitions of consistency have also been used in Continual Learning: Bhat et al. (2022a) and Bhat et al. (2022b). In the first one, the author proposes to add a regularization term that minimizes the $L_p$ norm between representations of a pair of samples. In the second, the authors proposes a self-supervised learning strategy to consolidate the knowledge of different tasks. However, both definitions of learning consistency differ strongly from the one used in this work.

## 5.1 LEARNING CONSISTENCY

Learning Consistency and the C-Score (Jiang et al., 2021) come from a line of work analyzing deep neural network training dynamics. One landmark study (Zhang et al., 2019) showed that deep neural networks had the capacity to learn even random noise. Later studies (Arplt et al., 2017), showed that natural images were learned faster than noise. Others analyzed how examples are forgotten during training (Toneva et al., 2019). Other metrics have been proposed for measuring learning dynamics such as model confidence, learning speed, holdout retraining and ensemble agreement (Carlini et al., 2019) which correlate well with each other. Learning speed in particular has been shown to correlate well with C-Score. Finally, a recent alternative for understanding per sample difficulty from the model's perspective is to measure the prediction depth in which a sample is correctly predicted at (Baldock et al., 2021).

## 6 CONCLUSIONS AND FUTURE WORK

In this work, we have analyzed how a memory population criterion based on learning consistency affects Experience Replay. We find that using only the most consistent samples in the memory is useful solely when having a limited compute budget. Otherwise, using Means of Features to populate the memory remains a strong baseline. However, selecting elements randomly from a set of the most consistent elements - a procedure we named Consistency AWare Sampling (CAWS) - does help and outperforms all baselines. This method relies on an accurate estimation of a measure called C-Score that is non trivial to calculate and would usually be impractical for Continual Learning scenarios. We propose proxies based on previous works that are easy to calculate and show little deviation from using ground truth C-Scores and still outperform our presented baselines. These proxies are not only useful for Continual Learning but can also be used for creating C-Score proxies for Curriculum Learning. In future work, we would like to find ways to determine *a priori* what is the right sampling ratio for a given task and how it relates to the C-Score distribution of a given dataset.

## 7 REPRODUCIBILITY STATEMENT

We will make our code public upon acceptance, along with examples of how to run the different experiments.

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

# A    APPENDIX

## A.1    C-SCORE DISTRIBUTIONS

The distribution that follows the C-Score for the dataset used in the experiments can be found in Figure 8. As mentioned in the main text, each dataset follows a different distribution, CIFAR-10 and Tiny Imagenet are more skewed than CIFAR-100. This distribution affects the results obtained by the methods proposed in the paper.

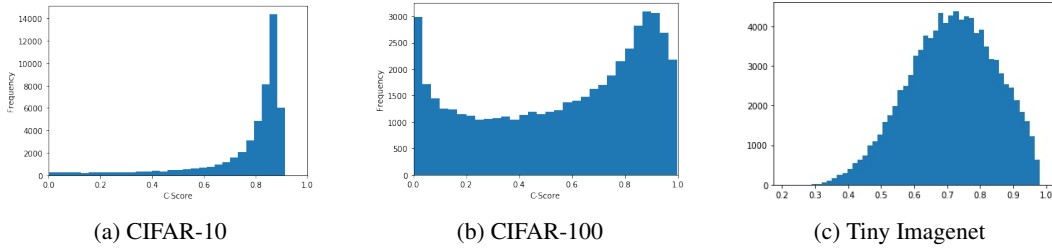

(a) CIFAR-10         (b) CIFAR-100         (c) Tiny Imagenet

Figure 8: C-Score distribution for MNIST, CIFAR-10 and CIFAR-100. We can see a clear difference between the different datasets, where the sets with the most data per class tend to cluster the C-Scores in the upper part, indicating that more examples are easy to train. On the other hand, CIFAR-100 shows a more uniform distribution.

## A.2    ALGORITHMS

The Algorithm 1 shows the details of CAWS. The only difference when applying the proxies, is that the Original C-Score is changed with the consistency approximation.

---

**Algorithm 1:** CAWS

**Components:**
- $D^t$: Dataset for task $t$.
- $M$: Memory.
- $N^c$: # of elements in memory of class $c$.
- $N^t$: # of elements to add per class.
- $\delta$: C-Score threshold.
- $C$: C-Score

**for** *classes in M* **do**
 $M \leftarrow$ remove $N^c - N^t$ elements
**end**
**for** *classes in $D^t$* **do**
 $x \leftarrow$ Sample $N^t$ from $D_c^t$ where $C(x) \geq \delta$
 $M.add(x)$
**end**
**Output:** Populated memory $M$.

---

## A.3    TABLE WITH RESULTS

Below you can see the details of all the results obtained. These are the same ones used to generate the figures in the main text.

Table 2: CIFAR10 - Split 5 - Memory size 50 - Model simple cnn

|  | # Epochs | 1 | 2 | 3 | 4 | 5 | 10 |
|---|---|---|---|---|---|---|---|
|  | Reservoir | 26.3% | 25.6% | 22.6% | 23.7% | 22.1% | 22.9% |
|  | Class Balance | 26.7% | 24.8% | 24.1% | 23.9% | 22.4% | 23.1% |
|  | Task Balance | 25.8% | 24.5% | 23.0% | 23.2% | 22.2% | 23.1% |
|  | Mean Features | 29.4% | 27.0% | 25.1% | 23.9% | 25.2% | 24.7% |
| Proxy 1 | C-score Lower | 20.5% | 19.3% | 19.7% | 18.8% | 19.4% | 19.4% |
| Proxy 1 | C-score upper | 28.4% | 28.8% | 27.9% | 27.0% | 26.3% | 25.9% |
| Proxy 1 | CAWS | 30.0% | 29.2% | 28.5% | 29.1% | 28.3% | 27.2% |
| Proxy 2 | C-score Lower | 19.9% | 19.7% | 19.8% | 19.6% | 19.6% | 20.0% |
| Proxy 2 | C-score upper | 28.3% | 27.9% | 27.2% | 26.1% | 26.3% | 25.4% |
| Proxy 2 | CAWS | 28.6% | 28.1% | 27.8% | 25.9% | 26.0% | 25.2% |
| Proxy 3 | C-score Lower | 18.6% | 19.1% | 19.0% | 18.6% | 18.8% | 18.4% |
| Proxy 3 | C-score upper | 28.3% | 26.8% | 27.0% | 25.8% | 25.0% | 25.3% |
| Proxy 3 | CAWS | 29.6% | 28.6% | 26.4% | 25.8% | 25.4% | 24.7% |
| No Proxy | CAWS | 29.5% | 27.9% | 26.0% | 25.7% | 25.1% | 24.9% |

Table 3: CIFAR10 - Split 5 - Memory size 100 - Model simple cnn

|  |  | 1 | 2 | 3 | 4 | 5 | 10 |
|---|---|---|---|---|---|---|---|
|  | Reservoir | 30.44% | 32.23% | 28.26% | 28.38% | 27.88% | 26.85% |
|  | Class Balance | 31.80% | 31.40% | 29.61% | 28.11% | 27.60% | 28.95% |
|  | Task Balance | 30.45% | 32.62% | 29.41% | 28.18% | 28.22% | 26.89% |
|  | Mean Features | 34.09% | 34.92% | 32.70% | 31.31% | 31.06% | 29.42% |
| Proxy 1 | C-score Lower | 22.99% | 23.12% | 22.89% | 21.78% | 21.98% | 20.64% |
| Proxy 1 | C-score upper | 31.23% | 32.98% | 31.69% | 30.96% | 30.28% | 29.14% |
| Proxy 1 | CAWS | 33.62% | 34.73% | 34.09% | 33.68% | 33.43% | 31.80% |
| No Proxy | CAWS | 32.45% | 34.22% | 31.46% | 31.52% | 31.75% | 30.01% |

Table 4: CIFAR10 - Split 5 - Memory size 250 - Model simple cnn

|  |  | 1 | 2 | 3 | 4 | 5 | 10 |
|---|---|---|---|---|---|---|---|
|  | Reservoir | 34.80% | 39.99% | 40.32% | 39.30% | 38.82% | 35.00% |
|  | Class Balance | 33.39% | 40.30% | 40.00% | 38.61% | 38.51% | 36.91% |
|  | Task Balance | 34.56% | 38.48% | 39.66% | 38.73% | 38.79% | 35.47% |
|  | Mean Features | 35.35% | 42.43% | 43.34% | 41.92% | 41.97% | 37.95% |
| Proxy 1 | C-score Lower | 24.87% | 30.34% | 31.04% | 29.26% | 29.09% | 27.48% |
| Proxy 1 | C-score upper | 36.25% | 38.93% | 40.22% | 40.00% | 40.17% | 38.31% |
| Proxy 1 | CAWS | 37.51% | 41.61% | 42.16% | 42.89% | 43.54% | 40.46% |
| No Proxy | CAWS | 37.60% | 41.31% | 42.11% | 41.77% | 40.94% | 39.71% |

Table 5: CIFAR10 - Split 5 - Memory size 500 - Model simple cnn

|  | # Epochs | 1 | 2 | 3 | 4 | 5 | 10 |
|---|---|---|---|---|---|---|---|
|  | Reservoir | 35.0% | 42.2% | 44.7% | 46.5% | 46.6% | 46.2% |
|  | Class Balance | 35.5% | 41.7% | 45.2% | 46.1% | 47.7% | 46.1% |
|  | Task Balance | 35.2% | 42.6% | 44.8% | 46.4% | 47.3% | 46.2% |
|  | Mean Features | 35.8% | 43.0% | 46.8% | 48.5% | 48.9% | 50.2% |
| Proxy 1 | C-score Lower | 25.0% | 31.2% | 34.5% | 35.0% | 34.0% | 32.7% |
| Proxy 1 | C-score upper | 39.0% | 42.3% | 45.1% | 46.7% | 47.4% | 46.5% |
| Proxy 1 | CAWS | 39.6% | 44.3% | 47.1% | 48.6% | 49.0% | 49.6% |
| Proxy 2 | C-score Lower | 21.7% | 25.1% | 27.3% | 28.8% | 29.1% | 29.3% |
| Proxy 2 | C-score upper | 37.8% | 42.5% | 45.0% | 46.2% | 47.9% | 47.3% |
| Proxy 2 | CAWS | 37.9% | 42.7% | 46.2% | 48.0% | 50.0% | 49.5% |
| Proxy 3 | C-score Lower | 19.7% | 21.5% | 21.8% | 22.5% | 23.0% | 23.3% |
| Proxy 3 | C-score upper | 37.7% | 41.4% | 44.7% | 45.0% | 45.9% | 47.3% |
| Proxy 3 | CAWS | 38.4% | 43.2% | 46.2% | 47.4% | 48.4% | 48.9% |
| No Proxy | CAWS | 39.0% | 44.1% | 46.9% | 48.7% | 49.7% | 49.8% |

Table 6: CIFAR100 - Split 5 - Memory size 500 - Model simple cnn

|         | # Epochs      | 1     | 2     | 3     | 4     | 5     | 10    |
|---------|---------------|-------|-------|-------|-------|-------|-------|
|         | Reservoir     | 11.0% | 13.3% | 14.7% | 14.8% | 14.9% | 15.6% |
|         | Class Balance | 11.5% | 13.2% | 14.8% | 14.5% | 14.9% | 15.9% |
|         | Task Balance  | 11.0% | 13.3% | 14.4% | 14.8% | 14.9% | 15.6% |
|         | Mean Features | 12.2% | 15.3% | 15.5% | 16.5% | 16.4% | 17.7% |
| Proxy 1 | C-score Lower | 8.7%  | 10.5% | 11.1% | 12.0% | 11.9% | 13.0% |
| Proxy 1 | C-score upper | 14.2% | 15.9% | 17.1% | 17.6% | 17.8% | 18.7% |
| Proxy 1 | CAWS          | 14.5% | 16.3% | 17.5% | 17.8% | 18.2% | 19.0% |
| Proxy 2 | C-score Lower | 8.1%  | 10.1% | 10.6% | 11.2% | 11.6% | 12.6% |
| Proxy 2 | C-score upper | 13.7% | 16.1% | 17.1% | 17.4% | 18.2% | 18.7% |
| Proxy 2 | CAWS          | 13.9% | 16.5% | 17.4% | 17.9% | 18.2% | 18.9% |
| Proxy 3 | C-score Lower | 9.1%  | 10.5% | 11.2% | 11.6% | 11.9% | 13.0% |
| Proxy 3 | C-score upper | 13.5% | 15.6% | 16.6% | 17.3% | 18.0% | 19.0% |
| Proxy 3 | CAWS          | 13.7% | 16.0% | 17.4% | 17.7% | 18.2% | 19.1% |
| No Proxy | CAWS         | 14.7% | 17.3% | 18.1% | 18.2% | 19.0% | 19.8% |

Table 7: CIFAR100 - Split 5 - Memory size 1000 - Model simple cnn

|         | # Epochs      | 1     | 2     | 3     | 4     | 5     | 10    |
|---------|---------------|-------|-------|-------|-------|-------|-------|
|         | Reservoir     | 11.1% | 15.5% | 17.8% | 18.3% | 18.7% | 19.3% |
|         | Class Balance | 11.5% | 15.3% | 17.9% | 18.2% | 18.3% | 18.7% |
|         | Task Balance  | 11.5% | 15.2% | 17.5% | 18.4% | 18.5% | 19.3% |
|         | Mean Features | 11.7% | 16.9% | 19.6% | 20.9% | 21.1% | 20.6% |
| Proxy 1 | C-score Lower | 8.7%  | 11.6% | 13.1% | 13.5% | 14.4% | 14.5% |
| Proxy 1 | C-score upper | 15.4% | 18.2% | 19.8% | 21.2% | 21.5% | 21.8% |
| Proxy 1 | CAWS          | 15.5% | 18.6% | 20.6% | 21.8% | 21.9% | 22.4% |
| Proxy 2 | C-score Lower | 7.6%  | 10.7% | 11.8% | 12.6% | 13.0% | 13.3% |
| Proxy 2 | C-score upper | 14.2% | 17.8% | 19.7% | 21.0% | 21.1% | 22.1% |
| Proxy 2 | CAWS          | 14.5% | 18.4% | 20.2% | 21.2% | 21.7% | 22.7% |
| Proxy 3 | C-score Lower | 8.8%  | 11.5% | 12.9% | 13.3% | 13.8% | 14.3% |
| Proxy 3 | C-score upper | 14.4% | 17.2% | 18.6% | 19.8% | 20.7% | 22.1% |
| Proxy 3 | CAWS          | 14.3% | 17.5% | 19.8% | 21.1% | 21.0% | 22.4% |
| No Proxy | CAWS         | 15.4% | 19.5% | 20.8% | 22.4% | 22.4% | 22.8% |

Table 8: CIFAR100 - Split 5 - Memory size 500 - Model Resnet

|         | # Epochs      | 1     | 2     | 3     | 4     | 5     | 10    |
|---------|---------------|-------|-------|-------|-------|-------|-------|
|         | Reservoir     | 15.4% | 19.1% | 21.3% | 21.8% | 22.8% | 25.5% |
|         | Class Balance | 15.7% | 19.3% | 21.1% | 22.1% | 23.1% | 26.0% |
|         | Task Balance  | 15.0% | 19.2% | 21.6% | 22.0% | 22.9% | 25.5% |
|         | Mean Features | 16.3% | 21.0% | 22.8% | 24.2% | 25.5% | 28.0% |
| No Proxy | CAWS         | 19.8% | 23.7% | 25.8% | 26.9% | 27.8% | 30.1% |

Table 9: CIFAR100 - Split 5 - Memory size 1000 - Model Resnet

|         | # Epochs      | 1     | 2     | 3     | 4     | 5     | 10    |
|---------|---------------|-------|-------|-------|-------|-------|-------|
|         | Reservoir     | 15.3% | 24.3% | 26.2% | 28.4% | 29.1% | 31.7% |
|         | Class Balance | 17.5% | 24.1% | 26.5% | 27.7% | 29.6% | 30.8% |
|         | Task Balance  | 15.3% | 22.3% | 25.6% | 27.7% | 29.1% | 31.3% |
|         | Mean Features | 17.9% | 25.4% | 29.3% | 30.9% | 31.0% | 34.3% |
| No Proxy | CAWS         | 21.3% | 27.8% | 31.1% | 31.6% | 32.6% | 35.9% |

Table 10: Tiny Imagenet - Split 10 - Memory size 1000 - Model Resnet

|         | # Epochs      | 1     | 5     | 10    |
|---------|---------------|-------|-------|-------|
|         | Reservoir     | 9.2%  | 12.3% | 12.7% |
|         | Mean Features | 10.3% | 14.7% | 14.5% |
| Proxy 1 | C-score Lower | 9.1%  | 12.7% | 13.3% |
| Proxy 1 | C-score upper | 9.5%  | 13.2% | 13.3% |
| Proxy 1 | CAWS          | 9.8%  | 13.4% | 13.5% |
| Proxy 2 | C-score Lower | 8.8%  | 12.5% | 11.7% |
| Proxy 2 | C-score upper | 10.8% | 14.9% | 15.8% |
| Proxy 2 | CAWS          | 11.0% | 15.5% | 15.7% |
| Proxy 3 | C-score Lower | 9.0%  | 12.3% | 12.0% |
| Proxy 3 | C-score upper | 9.6%  | 14.5% | 15.0% |
| Proxy 3 | CAWS          | 9.9%  | 14.9% | 15.2% |
| No Proxy | CAWS         | 9.8%  | 13.0% | 13.3% |

Table 11: Tiny Imagenet - Split 20 - Memory size 1000 - Model Resnet

|         | # Epochs      | 1    | 5     | 10    |
|---------|---------------|------|-------|-------|
|         | Reservoir     | 7.7% | 9.7%  | 8.9%  |
|         | Mean Features | 8.3% | 11.9% | 11.2% |
| Proxy 1 | C-score Lower | 7.9% | 10.3% | 9.7%  |
| Proxy 1 | C-score upper | 7.4% | 9.8%  | 10.0% |
| Proxy 1 | CAWS          | 7.8% | 10.3% | 9.9%  |
| Proxy 2 | C-score Lower | 6.4% | 8.2%  | 8.1%  |
| Proxy 2 | C-score upper | 8.6% | 11.7% | 12.0% |
| Proxy 2 | CAWS          | 9.0% | 12.5% | 12.6% |
| Proxy 3 | C-score Lower | 6.6% | 8.6%  | 9.0%  |
| Proxy 3 | C-score upper | 7.7% | 11.7% | 11.5% |
| Proxy 3 | CAWS          | 8.0% | 11.8% | 12.2% |
| No Proxy | CAWS         | 7.6% | 10.3% | 10.0% |

Table 12: Tiny Imagenet - Split 10 - Memory size 2000 - Model Resnet

|          |               | 1      | 5      | 10     |
|----------|---------------|--------|--------|--------|
|          | Reservoir     | 9.39%  | 17.23% | 17.51% |
|          | Class Balance | 8.84%  | 17.82% | 17.24% |
|          | Task Balance  | 9.28%  | 17.89% | 17.31% |
|          | Mean Features | 10.50% | 20.25% | 19.91% |
| No Proxy | C-score Lower | 9.11%  | 17.69% | 17.58% |
| No Proxy | C-score upper | 8.96%  | 17.68% | 16.24% |
| No Proxy | CAWS          | 10.12% | 18.29% | 17.95% |

Table 13: Tiny Imagenet - Split 20 - Memory size 2000 - Model Resnet

|          |               | 1     | 5      | 10     |
|----------|---------------|-------|--------|--------|
|          | Reservoir     | 7.12% | 14.49% | 14.64% |
|          | Class Balance | 6.19% | 15.70% | 15.02% |
|          | Task Balance  | 7.01% | 14.79% | 13.79% |
|          | Mean Features | 7.79% | 17.79% | 16.45% |
| No Proxy | C-score Lower | 7.04% | 15.77% | 14.92% |
| No Proxy | C-score upper | 7.22% | 15.33% | 15.15% |
| No Proxy | CAWS          | 8.15% | 15.43% | 15.08% |

