# OpenReview forum: "Populating memory in Continual Learning with Consistency Aware Sampling"
_ICLR.cc/2023/Conference — Submitted to ICLR 2023_

### Official Review · Reviewer_Yna5 · 2022-10-20

**Confidence:** 4
**Correctness:** 3
**Technical Novelty And Significance:** 2
**Empirical Novelty And Significance:** 1
**Recommendation:** 3

**Clarity, Quality, Novelty And Reproducibility:**

The paper is clear and easy to follow. However, the novelty of the paper is incremental. It merely proposes another heuristic approach for selecting samples for experience replay. However, the strategy is not supported by theoretical justification. Experimental results do not demonstrate that the method leads to competitive performance against prior works. Hence, the novelty and contribution of this work are limited and do not offer an improved understanding of the sample selection strategies for continual learning.

**Strength And Weaknesses:**

Strengths:

1. The paper is clear and easy to follow.

2. Experiments on three major continual learning frameworks are provided.

Weaknesses:

1. The idea is incremental.

2. The approach is heuristic and without much theoretical justification.

3. Comparison against prior methods is extremely limited.

**Summary Of The Paper:**

The paper explores the idea of addressing catastrophic forgetting through rehearsal using a memory buffer. It studies strategies to select the most important samples for storage in a memory buffer. The core idea is to design a consistency score that ranks samples for storage according to how easy they are to learn and at the same time are representative of previous tasks. Experiments using three common continual learning benchmarks are performed to demonstrate that the proposed method is effective.

**Summary Of The Review:**

The idea of experience replay using selected samples is not new and many existing works have been proposed. The proposed idea is a heuristic idea without no theoretical justification. Experiments also do not demonstrate that the proposed method leads to state-of-the-art performance. Hence, the paper does not provide theoretical novelties or sufficient empirical results to demonstrate its effectiveness. In conclusion, it is not ready for publication.

---

> ### Author Response · Authors · 2022-11-11
> **Answers to your concerns**
>
> We appreciate the comments. Below are some answers to your concerns:
>
> - We believe our work is more than an incremental improvement from previous ones. In this study, there is more than just using the C-Score in CL, as we present the following contributions:
>    - Use the C-Score to start understanding and explaining why random methods work so well compared to others in different scenarios.
>    - Once we took the first steps in understanding CL methods, we proposed two attributes that should be present in memory. Moreover, we give empirical evidence for why these attributes are valuable.
>    - We found the need for these attributes by using the C-Score. Therefore, we propose novel populating strategies based on the C-Score. However, this idea can be extended to more general solutions.
>    - In addition, we believe it is essential to highlight the contribution of proxies. It is not only valuable for applying CAWS in CL. However, it can also help in Curriculum Learning since it is a much cheaper method of finding a score related to the difficulty of elements.
>
> - The comparisons are based on the most used methods to fill the memory in CL. Random methods work well in a broad spectrum of tasks, so we compare our proposal with them.
>    - As mentioned to reviewer **1H7t**, we evaluated using GSS. However, experiments showed low effectiveness in class incremental scenarios, showing lower results than Mean Features. Given the number of experiments and the limitations in hardware, we compare our proposal to this method, plus the random baselines.
>    - Also, Mean Features is a widely used method since it is the one proposed in iCarl.

---

### Official Review · Reviewer_gaRy · 2022-10-24

**Confidence:** 4
**Correctness:** 2
**Technical Novelty And Significance:** 3
**Empirical Novelty And Significance:** 2
**Recommendation:** 3

**Clarity, Quality, Novelty And Reproducibility:**

Although the problem setting is new, my main concern is the limited novelty of the proposed CAWS method.The proposed CAWS method, as illustrated in Section 3, simply sampling randomly from the top C-Score samples.

- The paper is not easy to follow because they split the method and the results into multiple pieces, and some references and definitions are not set close to their first appearance.
- How the dataset is splited into 5 tasks is not explained explicitly.
- Why the accuracy would decrease with more training epochs in CIFAR-10-100 in Figure 3？

**Details Of Ethics Concerns:**

No ethics corcerns.

**Strength And Weaknesses:**

Pros:
The paper focused on a very interesting scenario in continual learning. The problem setting of continual learning with efficient learning is well motivated. It is reasonable to apply C-score in evaluating the learnablity of a sample and this method could be extended to other areas including Curriculum Learning.

Cons:
1. The structure of the paper is not clear. The authors proposed to use C-Score to measure how learnable a specific sample can be concerning a set of models, and C-Score of some datasets have already be calculated. They mentioned they needed a proxy to evaluate the score for the other datasets, but the definition of the proxy appear after several sections. The writing order will put extra reading burden for the readers. Also the original paper appears in Section 4.1, while the C-Score definition appears in Section 2.

2. For the datasets we already have the C-Score computed by Jiang 2021, does this C-Score rely on the model architecture? If we switch to a new architecture, does that mean this C-Score does not work any more?

3. There are a few proxies proposed by Jiang 2021, what is the difference between the proxy in this paper and the proxies in Jiang 2021?

4. How to determine the threshold of high C-score and low C-score? How to determine the X in CAWS?

5. The results in Figure 4 a and Figure 4 b are leading to two different directions when we increase the threshold. Could you please explain why the 1000 memory capacities would increase with threshold = 0.9 in the CIFAR100 and all the other curves seens going downwards?

**Summary Of The Paper:**

This paper investigates continual learning under limited memory setup  and proposes a storage policye CAWS (Consistency AWare Sampling), that leverages a learning consistency score (C-Score) to enable effective and efficient learning. CAWS adds diversity to the memory while allowing the model to learn a far more detailed decision boundary. Practical proxies that require no extra training can still achieve similar results according to some experimental results.

**Summary Of The Review:**

Overall, even though the problem is important, I find the structure of the paper needs further modifications. The method CAWS they proposed seems like just sampling from the top C-Score samples under the continual learning framework, I think the authors could put more emphasize on the novelty of this method and how much of the memory size is saved or the performance boost in using CAWS sampling.
On top of this, there are a few technical aspects (inconsistent results, hyperparameter thresholds etc) I do not completely agree with, therefore, would request authors to answer above questions for clarity.

---

> ### Author Response · Authors · 2022-11-11
> **Clarification and new references**
>
> We appreciate the comments. Below are some answers to the concerns:
>
> - Given a similar answer to reviewer **1H7t**. We structure the paper in this way to relate it to our work's 3 motivations and contributions. Here is a brief explanation of the structure:
>    - Introduction, where we motivate the idea of ​​having two essential attributes in our memory examples: Fast Learning and diverse samples. As an introduction, we do not go into details.
>    - Section 2: We start with the first attribute (Fast Learners) because, as shown in Table 1, previous methods fail to learn with few epochs. In this section, we explain the concept of this attribute and provide a possible solution (High-C).
>    - Section 3: Given one of the limitations found in High-C, we encourage ​​adding diversity (the second attribute). This section presents our second contribution (CAWS) with its results.
>    - Section 4: Given CAWS's limitations in current CL scenarios, we present alternatives to these. Here we present our third contribution (The proxies of the C-score).
>
> - C-Scores calculated on Jiang 2021 used thousands of models on different architectures. So while it may be architecture dependent, it should give a reasonable estimate. Moreover, some of our proxies can be calculated on the fly for any given architecture. Finally, there are works showing that examples are learned mostly in the same order [1][2].
>
> - They are actually quite similar. The main difference is that Jiang2021 never attempt to use pretrained models for calculating their proxies (i.e. they always have to train the model on the data to calculate those proxies), while our version of the proxies does not require this because we either use a pretrained on ImageNet model or because we use the current state of the model to calculate the proxy.
>
> - High-C and Low-C do not have a threshold. These methods select the K examples of each class with the highest (or lowest) C-Score, where K is the memory capacity for each class.
>
> - As a hyper-parameter, it is passed to the method. In section 3.2, we do an ablation study. We conclude that this value highly depends on the dataset's distribution of easy-to-learn samples.
>
> - The first phenomenon can be explained similarly to figure 1.c. Since the memory is small and we only select elements with a high C-Score, what happens is that we only represent a small spectrum of each class, decreasing the representativeness of the memory.
>    - Particularly in CIFAR10. As shown in Figure 7.a, there are no samples with C-Score over 0.95, which means this phenomenon is strongly reflected in this dataset.
>    - A similar effect occurs in CIFAR100. Since the memory is small, the method can only represent a few concepts. If we increase the threshold, diversity increases, but concepts are still not well represented, so we lose the ability to quickly recover from forgetting.
>    - It is not enough to have only one of the attributes. For a reliable memory, we need to represent easy-to-learn samples correctly and give diversity to the represented concept.
>
> - As it is common in CL, the split of each dataset is done randomly. One of the reasons for using multiple seeds is that each one performs a different split, helping to make the results more representative.
> - The main reason is that the model cannot correctly represent previous tasks with a minimal number of examples per class (only 10). For this reason, the weight modifications increase every time we increase the number of epochs, causing increased forgetting and, therefore, lower accuracy on previous tasks.
>
>
> [1] Hacohen, G., Choshen, L., & Weinshall, D. (2020). Let’s agree to agree: Neural networks share classification order on real datasets. 37th International Conference on Machine Learning, ICML 2020, PartF16814, 3908–3918.
>
> [2] Pliushch, I., Mundt, M., Lupp, N., & Ramesh, V. (2021). When Deep Classifiers Agree: Analyzing Correlations between Learning Order and Image Statistics. http://arxiv.org/abs/2105.08997

---

### Official Review · Reviewer_1H7t · 2022-10-25

**Confidence:** 5
**Correctness:** 2
**Technical Novelty And Significance:** 2
**Empirical Novelty And Significance:** 2
**Recommendation:** 3

**Clarity, Quality, Novelty And Reproducibility:**

The overall clarity and quality of the paper do not meet the ICLR standards.
The novelty of the paper is quite limited.
The reproducibility is not ensured since the codebase will be public after acceptance.

**Strength And Weaknesses:**

The main strength of the paper is the topic itself, selective buffering in continual learning.

However, the weaknesses of the paper overweigh the strength by a large margin:
- The organization of the paper is scattered and hard to follow. Interleaving methods and experiments are not clear. Moreover, the writing style makes it hard to distinguish the motivation and original parts of the paper.
- Saving the "easy to learn" examples in the buffer is counter-intuitive. An easy-to-learn example should already been learned by the model in early iterations, why bother saving it in the replay buffer instead of giving the model more exposure to hard examples?
- Lack of latest competing methods. As far as I am concerned, there has been recent work that studies the same topic as this paper, including but not limited to GSS [1], LARS[2] and DDR [3]. The authors should definitely compare the proposed method against these recent methods.


[1] Aljundi, Rahaf, et al. "Gradient based sample selection for online continual learning."NeurIPS 2019.

[2] Buzzega, Pietro, et al. "Rethinking experience replay: a bag of tricks for continual learning." ICPR 2021.

[3] Wang, Zifeng, et al. "SparCL: Sparse Continual Learning on the Edge." NeurIPS 2022.

**Summary Of The Paper:**

The paper presents an idea to select important examples to be buffered in continual learning. The proposed Consistency Aware Sampling (CAWS) considers examples that are easy to learn and representative of previous tasks. Empirical investigation has been conducted to evaluate the effectiveness of the proposed method.

**Summary Of The Review:**

Although the topic of selective buffering is interesting, the proposed method is not well justified neither intuitively or empirically. Therefore, I recommend rejection of the paper.

---

> ### Author Response · Authors · 2022-11-11
> **Improve text clarity and comparisons**
>
> We appreciate the comments. Below are answers to your concerns:
>
> - We structure the paper in this way to relate it to our work's 3 motivations and contributions. Here is a brief explanation of the structure:
>    - Introduction, where we motivate the idea of ​​having two essential attributes in our memory examples: Fast Learning and diverse samples. As an introduction, we do not go into details.
>    - Section 2: We start with the first attribute (Fast Learners) because, as shown in Table 1, previous methods fail to learn with few epochs. In this section, we explain the concept of this attribute and provide a possible solution (High-C).
>    - Section 3: Given one of the limitations found in High-C, we encourage ​​adding diversity (the second attribute). This section presents our second contribution (CAWS) with its results.
>    - Given CAWS's limitations in current CL scenarios, we present alternatives to these. Here we present our third contribution (The proxies of the C-score).
>
> - We believe this structure helps tell the story of how the research unfolded, as it would be highly unintuitive to just present our proxies without justification from the start.
>
> - We agree with the reviewer that easy-to-learn samples are examples that the model has already learned. However, the memory's motivation is to represent the distribution of the previous task so the model will not forget.
>    - Based on the definition of the C-Score, these samples represent the current distribution better. These samples are most consistent with the current scenario. As it is also shown in our work, populating the memory with just the harder examples works poorly (low-c). Harder-to-learn examples may be noisy or poorly represent the distribution of the task.
>    - We agree that the representation should include easy and not-so-easy-to-learn samples. For this reason, we include the second attribute (diversity).
>    - This idea is supported by the motivation of representativeness that the C-Score offers, along with the results.
>
> - We weren't aware of the paper [3], which will be published soon in NeurIPS. We appreciate it for letting us know, and it should be read and included in future work discussions.
>    - Concerning GSS, this approach was proposed for an online scenario. However, we tested it at the beginning of our experiments by using the implementation of Avalanche. We tested it since we were especially interested in domains with a low number of epochs. However, we found that GSS does not perform well in class incremental learning, so we decided not to continue running experiments with it, as we preferred to compare ourselves with Mean Features. In case you're interested, the average results that we obtained in CIFAR10 were:
> | GSS    | Memory Size |       |       |       |
> |--------|------------:|:-----:|:-----:|:-----:|
> | Epochs |         100 |   250 |   500 | 1,000 |
> |      1 |    23.80    | 27.70 | 31.50 | 30.20 |
> |     10 |    20.60    | 22.60 | 25.60 | 30.60 |
> |     20 |    20.80    | 23.80 | 25.30 | 28.40 |
>    - Columns are different memory size, and rows are results with different amountof epochs per task
>    - Concerning [2], here the authors present 5 tricks to improve performance of replay-based methods. 4 of those tricks can be applied to any method, including ours. The last trick is, as the author mention, an approximation of the gradient selection sample (GSS)
>    - For these reasons, we decided to compare our proposal with the selected methods.
>
> - It was our intention to make the code public when it was accepted, but we uploaded it as complementary material for the reviewers to check the reproducibility.

---

### Official Review · Reviewer_zPPV · 2022-11-02

**Confidence:** 4
**Correctness:** 3
**Technical Novelty And Significance:** 3
**Empirical Novelty And Significance:** 2
**Recommendation:** 6

**Clarity, Quality, Novelty And Reproducibility:**

The paper is quite clear and the results are interesting. The novelty is limited since C-score were already applied in CL, and the proxies are relatively straightforward. The authors included code for their experiments so reproducibility should not be an issue.

**Strength And Weaknesses:**

The strong points of the method are its conceptual simplicity (C-scores are an intuitive metric of relevance) and good performance compared to other memory-based baselines. The motivation section is especially insightful and helps to highlight how the memory selection problem is still very open and hard in general. The algorithm exposition is clear and its most relevant hyperparameter are ablated. The experimental section covers multiple datasets.

However the work also has some clear weaknesses
- One of the main contributions is the derivation of new proxies for C-scores, but the approximation accuracy (e.g. proxy C-score/actual C-score) of these proxies w.r.t. the real score or Jiang et al.'s approximation. The only metric reported is downstream accuracy of the classification task, but without checking the quality of the C-score approximation it is possible that the proxies are injecting additional biases that could actually be helpful for these specific tasks.
- The memory budgets considered in the experimental section are quite small (i.e. ~100-1000), and much smaller than what off-the-shelf hardware is capable of handling currently (i.e. 10k and up). It would be interesting to see how the various proxies and the baseline behave with a larger memory, and if the gain are preserved.
- For the real algorithm (i.e. the one using C-score proxies) only accuracy is reported, while usually the more relevant metrics in CL are forgetting or some sort of transfer.
- None of the experiments are reporting error bars or uncertainty, and it is not clear if the experiments with proxies have also been run with 3 seeds.

Minor comments:
- The baselines considered are only memory-based. While this is acceptable (CL is a broad field) it is a bit of a limitation of the comparison.
- The paper seems to only consider the setting where sample are explicitely labeled with their tasks during training. This is acceptable but it should be stated that it is a special case of the more general CL problem without explicit tasks or task boundaries.
- Accuracy and forgetting are never defined as metrics in the paper.
- In A.2 CAWS should include points with C(x) >= delta

**Summary Of The Paper:**

The paper proposes to use C-scores as a score to select representative samples to insert into a memory buffer in continual learning. More concretely the authors:
- describe current limitations of memory selection methods
- propose to use C-score, an expensive but accurate score capable to identify learnable samples, to select a memory that is highly representative of the class and can be efficiently learned
- identify two limitation in a naive use of C-scores and try to alleviate them:
    - C-scores are expensive to compute, and the authors propose more efficient proxies to approximate them
    - selecting a memory greedily w.r.t. C-scores can result in lack of diversity, so the author propose to select uniformly at random (promotes diversity) but only out of samples that achieve a C-score above a certain threshold (promotes relevancy)

The proposed method is evaluated using several ablations and baselines, but only within the context of other memory-based CL methods.

**Summary Of The Review:**

Overall the paper presents an interesting approach and is overall acceptable, but the empirical evaluation undermine the message that C-scores are an interesting metric in CL. In particular
- the quality of the proposed proxies are never truly investigated
- it is not clear which computational regime the authors are targeting for their approach, and they should justify their choiches for memory size, number of epochs, and size of datasets in each epoch in terms of memory and compute, providing back-of-the-envelope estimations supporting their choice (e.g. if this approach should target compute on a phone, a consumer desktop, an industry server, or a whole cluster).

---

> ### Author Response · Authors · 2022-11-11
> **Response**
>
> We appreciate the detailed and helpful comments. In the following paragraph, we answer some of the doubts and concerns:
>
> - Indeed, the only way we checked the effectiveness of the proxies relative to the original score is to measure the accuracy of the classification tasks. Since we are proposing a storage policy for Continual Learning, the most direct way to check effectiveness is the same scenario.
>    - However, since the proxies and the C-Score are calculated differently, we found that the ordering of the samples is different.
>    - An intuition of the positive bias is as follows: Since one of the goals of CAWS is to add intra-class diversity by highlighting the consistency of samples within a neighborhood, we can find different clusters of samples consistent within the same class. Our metric is further diversified by these multiple clusters.
>
> - The memory sizes used in the experiments are those commonly used in various CL works. We agree that no well-defined set of capacities or clearly defined scenarios and regimes exist in Continual Learning. However, we run several experiments with various memory capacities to prove one of our contributions (Taking a step toward understanding how memory should work in different Memory-based scenarios).
>    - Concerning a memory capacity of 10K or higher, the main problem is that the benchmarks used have around 5000 examples in each task. Saving all the examples of the first tasks did not seem appropriate to us.
>    - However, we do test a high memory capacity in Figure 1.a and 1.b, just to test the hypothesis that High-C lacked diversity.
>
> - We have the forgetting results for all experiments. We can include a few of these results in the document's final version. Adding the forgetting for the last experiment is indeed a good idea. The rest of the results can be added to the appendix.
>
> - Concerning Figure 6 (the only one without an error bar), those experiments were run with 3 different seeds. However, visually it did not provide any extra information and made the figure less clear, so we decided to remove them. However, it can be added, or we can add a comment on the figure caption to make this clearer.
>
> - Considering that one of our motivations is to take a step towards understanding what the best way to populate memory is, we decide to compare our approach only with direct baselines, which are the ones based on memory.
>
> - Section 2.1 indicates that our experiments and methods only consider task-label (or task-id) during training, which is standard in class-incremental learning scenarios.
>
> - We can add the definitions of Accuracy and Forgetting to the final version. It should follow similar definitions mentioned in Lopez-Paz & Ranzato (2017)
>
> - Thank you for bringing this error to our attention. We will fix it in the updated version.
>
> - Concerning the C-Score's lack of novelty in CL and the relatively straightforward proxy. We can add:
>    - We are unaware of any Continual Learning works that use the C-Score.  We would appreciate any related work that we might have missed. We know that few works mention the idea of using the consistency score, but none have directly applied the idea.
>    - We appreciate having a straightforward idea that works better than previous approaches.

---

### Decision · Program_Chairs · 2023-01-20

**Decision:**

Reject

**Justification For Why Not Higher Score:**

Only one reviewer voted for acceptance and he did not try to sway other reviewers.

**Justification For Why Not Lower Score:**

N/A

**Metareview: Summary, Strengths And Weaknesses:**

All reviewers agreed that the novelty of this work was limited and the proposed method lacked a theoretical justification. Reviewers also found the experimental validation and ablation studies to provide limited insights in the practical trade-offs. Reviewers appreciated the author's rebuttal as well as the work done to ensure results are reproducible, but overall concluded that this work was not ready for publication.


**Summary Of Ac-Reviewer Meeting:**

N/A